# A New Approach to Discovering HIV Symptom and Patient Clusters Using CNICS Data and Topic Modeling

Tru Cao
Biostatistics and Data Science
UTHealth Houston
Houston, USA
tru.cao@uth.tmc.edu

Weilu Zhao
Biostatistics and Data Science
UTHealth Houston
Houston, USA
weilu.zhao@uth.tmc.edu

Hulin Wu
Biostatistics and Data Science
UTHealth Houston
Houston, USA
hulin.wu@uth.tmc.edu

Thomas Giordano
Department of Medicine
Baylor College of Medicine
Houston, USA
tpg@bcm.edu

Maile Karris
Vc-health Sciences-schools
UC San Diego
La Jolla, USA
m1young@ucsd.edu

Sonia Napravnik
Division of Infectious Diseases
UNC-Chapel Hill
Chapel Hill, USA
sonia_napravnik@med.unc.edu

Meagan Whisenant
Department of Behavioral Science
MD Anderson Cancer Center
Houston, USA
mswhisenant@mdanderson.org

Veronica Brady
Cizik School of Nursing
UTHealth Houston
Houston, USA
veronica.j.brady@uth.tmc.edu

Greer Burkholder
School of Medicine
University of Alabama
Birmingham, USA
gburkholder@uabmc.edu

Katerina Christopoulos
School of Medicine
UC San Francisco
San Francisco, USA
katerina.christopoulos@ucsf.edu

Mari Kitahata
School of Medicine
University of Washington
Seattle, USA
kitahata@uw.edu

Edward Cachay
Division of Infectious Disease
UC San Diego
La Jolla, USA
ecachay@ucsd.edu

Barbara Gripshover
UH Cleveland Medical Center
Case Western Reserve University
Cleveland, USA
gripshover.barb@clevelandactu.org

Heidi Crane
Department of Medicine
University of Washington
Seattle, USA
hcrane@uw.edu

Kenneth Mayer
The Fenway Institute
Harvard Medical School
Boston, USA
khmayer@gmail.com

Deana Agil
School of Medicine
UNC-Chapel Hill
Chapel Hill, USA
deana_agil@med.unc.edu

*Abstract*—The identification of symptom patterns and assessment of their impacts on relevant health outcomes are important to symptom management and caring for people living with HIV (PWH). Research on HIV symptom clusters has been hampered by small sample sizes, conventional statistical methods not adequately capturing the intricate relationships among symptoms, and not associating symptom patterns with health outcomes. In this study, we proposed a new approach leveraging and adapting the Latent Dirichlet Allocation (LDA) topic modeling method to discover latent symptom clusters in one of the largest cohorts of PWH in the United States (US), sourced from the Centers for AIDS Research Network of Integrated Clinical Systems. Based on the reduced symptom space, patient clusters were then derived and analyzed for time to virological failure. The results showed that LDA outperformed traditional symptom clustering methods in identifying clinically meaningful symptom clusters. It included a novel systemic inflammatory response cluster among PWH in the US as a significant prognostic marker of virological failure. Moreover, the uncovered patient clusters were significantly distinguished in experiencing virological failure and could be characterized by distinct symptom clusters. The findings suggested a strong association between symptom patterns and subsequent virological failure among PWH. The study demonstrated the power of topic modeling as a new direction in symptom research to reveal complex symptom patterns, toward development of personalized symptom management and targeted interventions to improve the life span and quality of life in PWH.

*Keywords—unsupervised learning, dimensionality reduction, latent Dirichlet allocation, survival analysis, symptom index*

## I. INTRODUCTION

Acquired immunodeficiency syndrome (AIDS) caused by human immunodeficiency virus (HIV) infection has been considered as one of the three deadliest infections in the world, along with tuberculosis and malaria [1]. Although the expected lifespan of people living with HIV (PWH) using efficacious antiretroviral therapy (ART) now approaches that of the general population, PWH experience a higher burden of psychological and physical symptoms. In the combination ART era, the focus has shifted from opportunistic infections and survival to symptom management [2]. For effective symptom management, determination of the symptom burden of PWH, identification of risk factors for severe symptoms, and assessment of the impact of symptoms on relevant health outcomes are important [3][4][5].

Symptom clusters are concurrent symptoms related to each other and not required to have the same etiology [6]. While it is clinically essential to identify and utilize symptom clusters as diagnostic entities that may predict and have an impact on health outcomes [7][8][9], symptom clustering may provide analytical and computational advantages. Importantly, using

symptom clusters instead of single symptoms for analyses reduces the number of features while still preserving major patterns, helping to reduce the computational complexity and remove redundant features to avoid model overfitting [10][11].

The ability to characterize symptom clusters in PWH allows clinicians to identify individuals who may be at high risk for specific symptom experiences and associated poor outcomes, targeting patients most at risk for proactive and personalized symptom management prior to and during HIV treatment [7][8][12][13]. For example, identifying specific symptoms clusters that predict virological failure enables healthcare providers to carefully attend to PWH and further investigate possible underlying causes, whether they be undiagnosed substance use, intermittently used ART, violence or neglect in the home, or others.

While there have been many studies on symptom clustering among cancer patients [8][9][12], research on symptom clustering among PWH is sporadic and limited. A systematic review in [13] found only 13 eligible articles, four of which were just about one HIV symptom index and its validation [14][15][16][17]. Moreover, previous studies had three main limitations. First, the historical work on HIV symptoms mostly relied on a small number of patients. To obtain consistent and generalizable results requires study of HIV symptoms in a large and diverse population. Second, most of the studies stopped at obtaining HIV symptom clusters and provided little information about whether and how symptom clusters may be associated with or predict health outcomes of PWH. Third, for HIV symptom clustering, only traditional statistical methods such as principal component analysis (PCA) [18] or hierarchical clustering analysis (HCA) [19] were used. The state-of-the-art machine learning methods with high performance and leveraged interpretability can help capture latent relationships between symptoms and PWH, to provide better characterization and prediction models of HIV symptom patterns.

In this paper, we proposed a new approach to symptom clustering that adapted Latent Dirichlet Allocation (LDA) from the text domain [20] to discover symptom clusters in a large cohort of PWH from the Centers for AIDS Research (CFAR) Network of Integrated Clinical Systems (CNICS), one of the largest integrated clinical data repositories for PWH in the United States (US) [21]. We then proposed to use $k$-means [22] to cluster patients represented as probabilistic distributions on the LDA-produced symptom clusters and utilize the PWH cluster membership for predicting time to virological failure. We found a new symptom cluster of systemic inflammatory response that was not detected or described in previous symptom studies on PWH populations in the US. The results also showed that LDA outperformed other dimensionality reduction and factor analysis methods in differentiating patient clusters regarding their association with virological failure. To our knowledge, this is the first study to employ the topic modeling approach for symptom and patient clustering to any diseases, including cancer and HIV.

The paper is organized as follows. Section II reviews the related work. Section III presents the CNICS PWH database,

data preprocessing, methods for symptom and patient clustering, and time to virological failure analysis. Section IV presents our study results in detail. Section V discusses the findings, their clinical implications, and limitations of this study. Finally, Section VI draws some concluding remarks.

## II. RELATED WORK

### A. HIV Symptom and Patient Clustering

Previous HIV symptom studies mainly used PCA for symptom clustering and HCA for patient clustering. The historical work resulted in symptom clusters with significant variances due to differences in the methods and symptom assessment tools used [23][24][25][26][27][28][29][30][31]. Applying confirmatory factor analysis (CFA) [32] to 246 PWH, [23] detected six symptom clusters, namely, malaise/fatigue, confusion/distress, fever/chills, gastrointestinal discomfort, shortness of breath, and nausea/vomiting. In [24] PCA was applied to 1,885 PWH in the US undesirably resulting in two main unnamed symptom factors that were highly overlapping and strongly correlated. Meanwhile, in another study PCA applied to 951 PWH with dual HIV and diabetes diagnoses showed changes in symptom patterns as compared to those with HIV only [25]. Outside PWH populations in the US, in [26] PCA was applied to 191 PWH in Jamaica identifying three clusters of depressive symptoms, namely, cognitive-affective, negative cognitions, and somatic symptoms. In [27] HCA was used and identified three patient subgroups of 2,505 PWH according to the number of bothersome symptoms reported in French hospitals. Another study on 1,116 PWH in China, also using PCA, identified five symptom clusters, including cognitive dysfunction, mood disturbance, wasting syndrome, dizziness/headache, and skin-muscle-joint disorder [28].

Recently, in a CNICS cohort of 2,000 PWH, PCA was used identifying four symptom clusters, namely, psychological, body image, gastrointestinal, and pain [29]. In contrast to symptom clustering only, other studies [30][31] applied HCA and identified five symptom-based patient subgroups among a few hundred PWH in South Africa and Uganda. Notably, except for [23] exploring if symptom clusters were signs of treatment failure, impacts of symptom clusters on heath outcomes were not analyzed in the other cited studies.

### B. Clustering Approaches

The main approach to symptom clustering is factor analysis, while that to patient clustering is cluster analysis [33][34][35]. HCA groups data directly on original features based only on their relative similarity and thus might not well capture latent relationships between the features. This drawback motivated development of dimensionality reduction methods such as PCA or LDA. PCA restrictively assumes that principal components are orthogonal as linear combinations of the original features and can take negative values that make data interpretation difficult [36]. Biclustering performs simultaneous row-column clustering that produces sub-matrices (i.e., biclusters) that satisfies some characteristics [37][38]. As such, biclustering requires the number of clusters on rows to be the same as the number of clusters on columns, which may not be practicable.

Moreover, biclustering is mostly based on value similarity and not well applicable to co-occurrence data [39].

In contrast, latent semantic analysis (LSA), the first topic modeling method, aims at discovering hidden topics in a set of documents, each of which is a vector over a set of words, and documents are then represented as vectors over those hidden topics [40]. As extensions of LSA, probabilistic LSA (pLSA) [41] and LDA offer a clear interpretation of a hidden topic being a probability distribution over a set of words and a document being a probability distribution over a set of hidden topics.

Although originated from the field of natural language processing, topic modeling has had many successful applications in bioinformatics, using analogy between the concept of document-topic-word and that of biomedical objects [42]. Recently, LDA was extended to model disease diagnoses in electronic health records (EHRs) that effectively identified hidden disease clusters and stratified patients into differentiable subgroups [43]. Among topic modeling methods, LDA as a generative probabilistic model and Non-negative Matrix Factorization (NMF) [44] as a non-probabilistic linear-algebra model are generally the most used and best performing methods [45][46]. Moreover, LDA offers superior probabilistic interpretability [47].

### C. Limitations of Previous Studies

As reviewed above, previous studies on HIV symptom clustering experienced three main limitations: (1) they were based on small PWH populations resulting in limited generalizability and inconsistent symptom clusters across studies; (2) they did not benefit from the state-of-the-art statistical and machine learning methods to identify consistent and clinically meaningful HIV symptom clusters in a large population of PWH; and (3) there was little research done and information found on how HIV symptom clusters may predict and impact health outcomes of PWH.

### III. METHODS

### A. Data Source

This study utilized the data from the CNICS database including PWH receiving care at ten participating academic medical centers across the country. The database collects comprehensive demographics, laboratory measurements, ART information, and patient-reported outcomes (PROs) [21]. The data collection process involves the integration of EHRs, administrative data, laboratory data, as well as PROs collected approximately every 6 months during routine clinical visits.

Symptoms are measured and reported using the HIV Symptom Index (HIV-SI) [48]. The HIV-SI is valid, reliable, and considered the gold-standard HIV symptom index for HIV symptom research [25][29]. The HIV-SI questionnaire contains 20 symptoms (e.g., anxiety, hand/foot pain, skin problems) for participants to report which symptoms are present and indicate the distress score for each symptom on a five-point Likert scale; 0 = symptom absence, 1 = *it doesn't bother me*, 2 = *it bothers me a little*; 3 = *it bothers me*, and 4 = *it bothers me a lot*. Besides its large size, another advantage of the CNICS PWH data is that the symptom distress scores provide a richer semantic space for analyses than many other PWH cohorts with only dichotomous symptom measures.

Our initial PWH study cohort included 17,302 unique patients and their total 80,262 PROs from 2005 to 2022; each patient could have multiple PROs over time. The demographic characteristics of this cohort were diverse, with a mean age of 47.5 (±12.2) years and majority of 85.3% male participants. The cohort included a significant representation of racial and ethnic minorities, with 32.1% African American and 15.9% Hispanic.

### B. Data Preprocessing

HIV ribonucleic acid (RNA) viral load data were used for time to virological failure analysis, with a threshold of 50 copies/mL considered indicative of viral suppression and two consecutive readings $\geq$ 50 copies/mL as virological failure [49]. We considered only the PWH with at least one RNA reading < 50 copies/mL (i.e., viral suppression). For each unique patient, we selected only one PRO that was closest and prior to the first viral suppression and included complete distress scores for all 20 symptoms. The final PWH cohort for this study included 12,983 PROs/unique PWH.

### C. Symptom Factor Analysis

In many fields of study, including healthcare and medical research, we often deal with high-dimensional data. For example, in the context of this study, each patient symptom profile is described over many single symptoms, each of which can be considered a separate dimension. While high-dimensional data can provide a wealth of information, it also presents several challenges that can make it difficult to visualize the data, identify patterns, and build predictive models [50].

Dimensionality reduction techniques are used to reduce the number of dimensions (i.e., features or variables) in a dataset while preserving as much of the relevant information as possible. The benefits of dimensionality reduction are manifold. Firstly, it can help to simplify the data, making it easier to explore and visualize. Secondly, it can help to remove noise and redundancy in the data, thereby improving the performance of predictive models. Thirdly, in the context of this study, dimensionality reduction can help to identify subgroups of patients with similar symptom profiles, which can provide symptom clusters as diagnostic entities and valuable insights for personalized treatment strategies.

The following methods were employed to reduce the dimensionality and identify principal symptom clusters from the PROs of the PWH study cohort: PCA [18], CFA [32], NMF [44][51], pLSA [41][52], and LDA [20][53]. We adapted pLSA and LDA from text topic modeling to discover latent symptom clusters within a cohort of PWH.

For LDA, we mapped its methodological concepts to those of symptoms and patients where *word-document matrix = symptom-patient matrix*, *words = symptoms*, *documents = patients*, and *topics = latent symptom clusters*. We constructed a symptom-patient matrix where each element is the distress score (from 0 to 4) of a symptom reported by a patient.

Symptom clusters were then derived by LDA, each of which is a probability distribution on the HIV-SI 20-symptom set.

The above-selected dimensionality reduction methods (i.e., PCA, CFA, NMF, pLSA, and LDA) were applied to the PWH study cohort. For each method, we explored various parameter settings for optimal performances. The number of symptom clusters and the representative symptoms in each cluster were determined based on model selection criteria. For PCA and CFA, symptoms with loadings $\geq$ 0.40 were considered meaningful [32]. For NMF, pLSA, and LDA, we employed the "top N words" method, i.e., selecting the representative symptoms with the highest weights and largely different from the remaining symptoms in each cluster. The symptom clustering results were validated by examining the clinical interpretability and relevance of the derived symptom clusters with their representative symptoms.

### D. Patient Cluster Analysis

After symptom factor analysis, we utilized the obtained symptom clusters to represent and cluster patients. Then the patient cluster membership was utilized in the time to virological failure analysis. Each patient was represented as a vector over the symptom clusters identified by a dimensionality reduction method. $k$-means was then employed for clustering the patients, as it works well with large datasets and is known for its high efficiency and performance [54]. We also applied $k$-means and biclustering to directly cluster the patients on the raw data without dimensionality reduction (i.e., with the 20 HIV-SI symptoms).

The elbow method was used to determine the reference number of clusters for each method first [55][56]. This point represents a balance between capturing the underlying structure of the data and avoiding overfitting. We then tuned the number of clusters around this reference number until obtaining the optimal number with respect to the analysis of discourse.

Finally, for all patient clustering methods, we evaluated the quality of the resulting patient clusters with respect to how well they are differentiated by heath outcomes such as time to virological failure in this study. Fig. 1 illustrates our proposed symptom score-based LDA framework for symptom and patient clustering, where each $d_{ij}$ is the distress score of symptom $s_i$ occurring in patient $p_j$ ($1 \leq i \leq m, 1 \leq j \leq n$).

### E. Virological Failure Analysis

To assess the prognostic value of the identified symptom and patient clusters and their potential clinical implications for PWH, we conducted a time to virological failure analysis using Kaplan-Meier curves and log-rank tests [57][58]. For each patient cluster, a Kaplan-Meier curve was generated with time to virological failure as the event of interest from the first viral suppression as time zero and censoring at the last available viral load measurement. To determine the statistical significance of differences in virological failure between patient clusters, we used both visual comparisons and log-rank tests. The log-rank test compares the observed number of virological failure events in each patient cluster to the expected number under the null hypothesis of no difference between clusters.

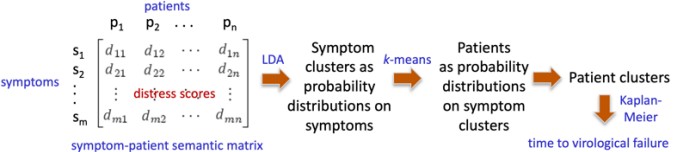

Fig. 1. The proposed LDA framework for symptom and patient clustering.

In addition, we performed pairwise log-rank tests between all clusters and adjusted for multiple comparisons using the Bonferroni correction [59]. These post-hoc tests provided a more detailed understanding of the relationships between individual symptom clusters and virological failure outcomes. Kaplan-Meier curves were generated using the lifelines and matplotlib modules in Python [60]. All statistical analyses were conducted using Python 3.11.8.

The more differentiated the patient clusters are, the better the underlying symptom clusters characterizing them could predict their respective health outcome event. This analysis could help identify patient subgroups at higher risk of virological failure, enabling targeted interventions and personalized care strategies to improve treatment outcomes and quality of life for PWH.

## IV. RESULTS

### A. Symptom Prevalence Summary

The symptom prevalence among 12,983 PWH in the study cohort is summarized in Table I. It shows the rank order of symptoms based on their occurrence rates, providing insights into the most common complaints among HIV patients. Fatigue was the most prevalent symptom, followed by Sleep Trouble and Muscle/Joint Pain. Other two symptoms with prevalence near to 50% are Anxiety and Sadness. Lower-ranked symptoms Weight Loss, Hair Loss, or Nausea despite being less frequent, still affect about a quarter of the PWH population, underscoring the diverse and complex symptomatology in PWH.

### B. Symptom and Patient Clusters

PCA, CFA, and NMF suggested four symptom clusters as the optimal solution, while pLSA and LDA identified five symptom clusters. We chose to illustrate the symptom clusters identified by LDA, for its superior performance in capturing the thematic structure of symptom co-occurrence patterns and identifying interpretable symptom clusters. To visualize the five symptom clusters detected by LDA, we created a heatmap shown in Fig. 2 where each row represents a symptom, each column represents a latent symptom cluster, and the color intensity indicates the occurrence probability of each symptom (cf. weights for other methods). This visualization allows us to recognize the co-occurrence patterns of symptoms and how they form distinct clusters.

As marked by the red boundaries in Fig. 2, the first symptom cluster consists of Sex Problems, Sadness, Anxiety, Fatigue, and Sleep Trouble, with their highest occurrence probabilities and a significant probability drop from the 6th ranked symptom. For the next cluster, the 4th ranked symptom Fatigue has an occurrence probability of 0.064, with a 0.106 difference from 0.17 of the least ranked symptom Weight Loss in the cluster.

The selection of the representative symptoms for each of the other symptom clusters was performed similarly. In addition, the representative symptoms were selected so that they did not overlap across different symptom clusters. Therefore, Hand/Foot Pain ranked 4th in the SIR cluster was assigned to the Pain cluster that it strongly belongs to.

The five LDA-based symptom clusters and their representative symptoms are presented in Table II. To label each symptom cluster, we chose a general term with a clinical meaning that is related to the included symptoms. For example, for Cluster 3, while diarrhea and gas or bloating in the stomach are obviously gastrointestinal symptoms, headaches were found to be associated with gastrointestinal disorders [61]. For Cluster 4, while fever is a body's natural response to inflammatory stimuli, inflammation could cause dizziness [62] and appetite loss could be associated with systemic inflammation [63]. We suggest further clinical investigation to find and confirm appropriate labels for such symptom clusters.

TABLE I. THE RANKED LIST OF SYMPTOM OCCURRENCES

| Rank by Occurrence | Symptom | Percentage (n = 12,983) |
|---|---|---|
| 1 | Fatigue | 58.65 |
| 2 | Sleep Trouble | 53.89 |
| 3 | Muscle/Joint Pain | 51.11 |
| 4 | Anxiety | 49.22 |
| 5 | Sadness | 49.05 |
| 6 | Hand/Foot Pain | 43.38 |
| 7 | Memory Loss | 43.05 |
| 8 | Body Image | 42.13 |
| 9 | Sex Problems | 38.75 |
| 10 | Headache | 38.64 |
| 11 | Bloating/Gas | 37.09 |
| 12 | Skin Problems | 34.88 |
| 13 | Diarrhea | 34.81 |
| 14 | Dizziness | 33.02 |
| 15 | Cough/SOB | 31.87 |
| 16 | Appetite Loss | 28.13 |
| 17 | Fever | 27.24 |
| 18 | Weight Loss | 25.76 |
| 19 | Hair Loss | 25.36 |
| 20 | Nausea | 22.53 |

Note: Patients reported more than one symptom, so the percentages do not add up to 100%.

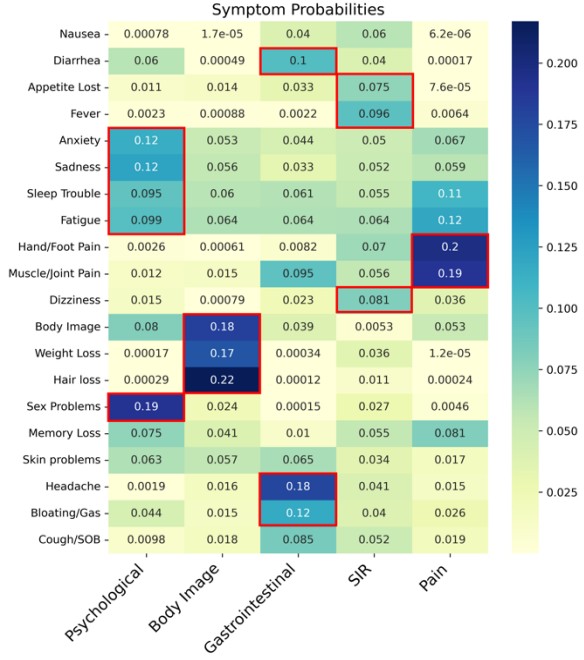

Fig. 2. The heatmap of LDA-based symptom clusters.

TABLE II. THE LDA-BASED LATENT SYMPTOM CLUSTERS

| Cluster | Representative Symptoms | Cluster Label |
|---|---|---|
| 1 | Sex Problems, Sadness, Anxiety, Fatigue, Sleep Trouble | Psychological |
| 2 | Hair Loss, Body Image, Weight Loss | Body Image |
| 3 | Headache, Bloating/Gas, Diarrhea | Gastrointestinal |
| 4 | Fever, Dizziness, Appetite Loss | Systemic Inflammatory Response (SIR) |
| 5 | Hand/Foot Pain, Muscle/Joint Pain | Pain |

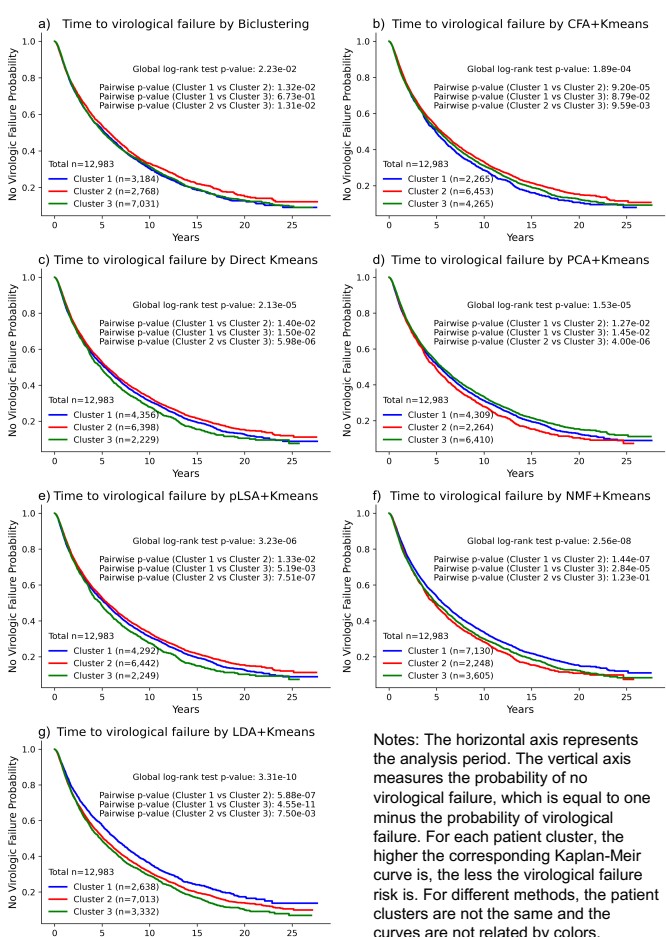

Fig. 3. The time to virological failure analysis based on the symptom and patient clusters produced by the seven methods.

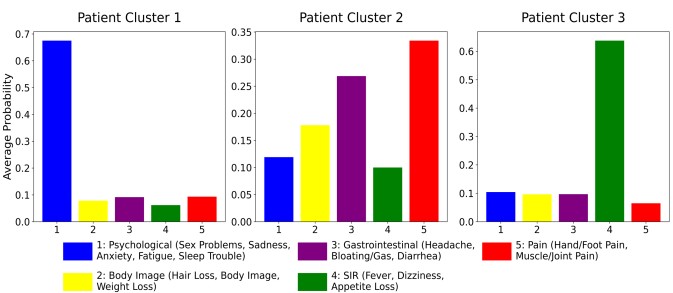

Fig. 4. LDA-based symptom characterization of the patient clusters.

The top and representative symptoms in each LDA-based cluster could be clinically interpreted and collectively characterized and labeled by a unique symptom profile. This demonstrated that LDA could capture latent co-occurring symptom patterns among PWH in this large study cohort. The

Psychological, Body Image, Gastrointestinal, and Pain clusters identified by LDA aligned with the symptom clusters reported in a recent study on a smaller CNICS cohort of 2,000 PWH [29]. This consistency with prior research supported the validity and clinical relevance of these clusters. Notably, the SIR cluster represents a novel finding in the context of HIV symptom clustering in PWH populations in the US. The presence of the SIR cluster suggested that LDA was able to uncover previously unrecognized symptom patterns that could have important implications for PWH symptom management and treatment.

For patient clustering using *k*-means after dimensionality reduction by PCA, CFA, NMF, or pLSA, or directly on the original 20-symptom space, the elbow plots suggested two, three, or four as the optimal numbers of patient clusters. In the case of LDA, the elbow plots indicated three, four, or five patient clusters as the most suitable options. For biclustering, the number of symptom clusters and the number of patient clusters must be the same. To verify the suggestions from the elbow method and determine the most appropriate numbers of clusters, we performed Kaplan-Meier analysis with the number of patient clusters ranging from two to six for all methods.

*C. Differentiation of Patient Clusters*

Survival curves, i.e., Kaplan-Meier plots, were generated to examine the separation between different patient clusters with respect to time to virological failure. Clusters with clearly separated survival curves suggest meaningful differences in clinical outcomes, thereby confirming the appropriateness of the obtained symptom and patient clusters.

Fig. 3 presents the time to virological failure for the patient clusters identified by each of the seven different methods, presented in the order from the largest to the smallest (i.e., the best) global rank-test *p*-values. LDA yielded the smallest global *p*-value of $3.31 \times 10^{-10}$, indicating a strong association between the identified patient clusters and virological failure. It was closely followed by NMF with a *p*-value of $2.56 \times 10^{-8}$. Further down there were pLSA, PCA, direct *k*-means, CFA, and biclustering. The result is consistent with the literature that LDA and NMF are among the best topic modeling methods [45][46]. In terms of pairwise rank-tests, the smallest *p*-values were also observed for the clusters identified by LDA.

The LDA-based patient clusters demonstrated well-separated Kaplan-Meier curves that did not cross, indicating a consistent association between symptom profiles and virological failure risk over time. In contrast, some of the curves for the patient clusters identified by the other methods showed crossing, implying unstable associations. These findings showed the superior performance of LDA in terms of the stability and prognostic value of the identified symptom and patient clusters. It underscored LDA's potential utility in identifying PWH at high risk of adverse outcomes based on their symptom patterns.

*D. Symptom Characterization of Patient Clusters*

To further characterize the patient clusters identified through *k*-means by the underlying symptom clusters produced by LDA, we examined the distribution of these symptom clusters in each patient cluster. The bar charts in Fig. 4 present a breakdown of the average probability weights of the symptom clusters in each patient cluster. The dominant symptom clusters in each distribution of the symptom clusters could characterize the corresponding patient cluster, offering insights into the symptom experiences of different patient clusters and their associated risks of virological failure. As shown, Patient Cluster 1 was clearly dominated by the Psychological symptom cluster, Patient Cluster 3 by the SIR symptom cluster, and Patient Cluster 2 by the remaining Pain, Gastrointestinal, and Body Image symptom clusters. Our analysis revealed that the patient subgroup with the highest risk of virological failure was predominantly characterized by the SIR symptom cluster.

V. DISCUSSION

*A. Topic Modeling Approach*

This study investigated topic modeling as a new approach to discover symptom patterns in PWH from the CNICS cohort. We adapted LDA in the text data domain for symptom clustering and evaluated its performance with traditional factor analysis methods such as PCA and NMF. The findings demonstrated that LDA outperformed the other methods in providing more interpretable and relevant symptom clusters and prognostically differentiable patient clusters with respect to the time to virological failure analysis. The results also proved the advantage of dimensionality reduction showing patient clustering based on latent symptom clusters could produce more distinguished patient clusters than on the original space of the 20 single symptoms, using the same *k*-means method. Last but not least, a topic modeling method like LDA is unsupervised, as opposed to supervised machine learning, and this is a notable advantage for not requiring intensive human labor to manually annotate data in EHRs.

*B. Implications for Clinical Practice*

Our results showed that LDA identified five distinct symptom clusters and three respective patient clusters that were associated with different risks of virological failure, highlighting the prognostic value of the discovered symptom clusters. The SIR cluster is a new finding and associated with the highest risk of virological failure. This cluster may represent a distinct subgroup of PWH who require additional monitoring and targeted interventions to prevent adverse outcomes.

On one hand, the association between symptom clusters and virological failure suggests that symptom profiles could be used as a screening tool to identify patients at high risk of treatment failure. This could inform decisions about the frequency of viral load monitoring, adherence support, and the need for regimen modifications [64][65]. On the other hand, this association may warrant further investigation to understand its clinical significance and potential mechanisms. For an example of the SIR symptom cluster, it could be due to the association of systemic inflammation and immune reconstitution inflammatory syndrome in PWH that results in rapid clinical deterioration [66].

The primary purpose of this work was to further define symptom experiences in PWH and explore the utility of

symptom clusters to predict health outcomes. By identifying clinically relevant symptom clusters, it aimed to inform personalized treatment strategies and potentially improve patient outcomes. Our overall long-term goal is to provide healthcare providers with early cues at the time of viral suppression, to reduce the burden of impacting symptoms, pursue early prevention of virological failure, and avoid adverse health outcomes.

## C. Limitations

Despite the novel methodological approach, insightful findings, and clinical significance of this study, it is important to acknowledge its limitations. The study relied on self-reported data, which might be subject to individual bias and differences in symptom perception. The influence of confounding factors like pre-treatment viral load, CD4 cell count, medication adherence, and comorbidities on symptom experiences was not addressed. We initiated our analysis at the first viral suppression to mitigate the impact of these important clinical factors. Examination of the impact of these factors will be the aim of future work. The cross-sectional nature of the study limits our ability to make causal inferences about the relationship between symptom clusters and virological failure risk.

## VI. Conclusion

The novelty and contribution of this study are threefold. First, it adapted LDA and demonstrated its potential as a robust tool for identifying clinically meaningful and prognostically relevant symptom clusters in PWH. The LDA-based approach outperformed other dimensionality reduction methods and direct clustering on the original symptom space in capturing the thematic structure of symptom co-occurrence patterns and identifying differentiable patient subgroups regarding time to virological failure. The identification of the five distinct symptom clusters, including a novel SIR cluster, provided a nuanced understanding of the symptom experiences of PWH in the US and underscored the prognostic value of these clusters. This study was the first and opened a new research direction in the use and further development of topic modeling techniques for symptom research. Second, it was also the first to utilize symptom-based patient cluster membership to predict health outcomes such as virologcal failure among PWH. Third, it was performed on one of the largest cohorts of PWH in the US that contained PROs using a standard symptom index, allowing stability and generalizability of obtained results.

The discovery of the SIR symptom cluster and its association with virological failure risk suggests further research into its mechanisms and implications for PWH symptom management. The integration of LDA-based symptom clusters with other clinical data could lead to the development of comprehensive risk prediction models and decision-support tools for personalized HIV care. The LDA-based symptom clustering approach could be extended to other chronic diseases, such as cancer, diabetes, and cardiovascular diseases, where the co-occurrence of multiple symptoms significantly impacts patient health outcomes and quality of life. Future research could further analyze risks of the obtained symptom-based patient clusters regarding confounding factors and other health outcomes.

## Acknowledgment

The data used in this study are provided by CNICS, an NIH funded program (R24 AI067039) made possible by the National Institute of Allergy and Infectious Diseases (NIAID). This work is partially supported by the Texas Developmental Center for AIDS Research (D-CFAR) funded by NIH P30 AI161943. Dr. Whisenant is supported by research career development award K12AR084228 (PI: Berenson), funded through NIH institutes NIAMS, NICHD, NIAID, and OD. The content is solely the responsibility of the authors and does not necessarily represent the official views of the National Institutes of Health.

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
