# OpenReview forum: "A New Approach to Discovering HIV Symptom and Patient Clusters Using CNICS Data and Topic Modeling"
_IEEE.org/EMBS/BHI/2024/Conference — IEEE BHI'24_

### Official Review · Reviewer_d3t4 · 2024-08-08
**Identification of symtom pattern clusters associated with health outcomes in PWH population using novel topic modeling**

**Overall Rating:** 7
**Confidence:** 5

**Other Quality Metrics:**

- Clarity of writing: Great
- Clinical Significance: Great
- Methodological Novelty: Good
- Experiments and Results: Good

**Questions For The Authors:**

- Page 3 Section B:
    - This section was a bit confusing. My understanding is as follows:
        - Only 1 PRO was selected per unique PWH → There were 12,983 PROs (unique PWH) after selection.
        - Each of the 12,983 PWHs experienced both virological failure and virological suppression.
        - When selecting PRO within each PWH, the PRO closest to and prior to (and still in virological failure status) virological suppression was selected.

        → Is this correct?

- The concept of time to virological failure and viral suppression is confusing from the text. A visualization with an example showing how time to virological failure is defined would be helpful.
- Similar comment here: Page 4 Section E
    - “Kaplan-Meier curves were generated with time to virological failure as the event of interest from the first viral suppression (i.e., RNA < 50 copies/mL) as time zero and censoring at the last available viral load measurement.” → This would be easier to understand with visuals.
    - An additional description of the Kaplan-Meier curve (how it should look, what each axis means, etc.) would help better understand the results in the later section.
- Figure 4:
    - This figure is too small.
    - The bar plots for clusters 4 and 5 are missing. Were they not showing strong evidence like other clusters?
    - The Kaplan-Meier curve here seems to be a duplicate of the one in Figure 3.
- In Discussion/Conclusion:
    - The discovery of the SIR cluster would be more convincing if there were supporting clinical studies that observed similar patterns.
- General questions: Were there any studies done in this area using machine learning algorithms? For example, using data points of symptoms to predict the time to virological failure with an ML model?
- Minor comment on format: For references in the text, if the reference numbers are consecutive, they could be written as [23-31] instead of [23][24][25][26][27][28][29][30][31] for better readability.
- Each figure should have a sub-figure legend with a description if it has multiple plots.

**Strengths:**

- The paper's topic is novel because there are limited studies on symptom clustering in the PWH population. Previous studies used traditional approaches like PCA, but not more advanced clustering methods.
- The paper is well-structured. The introduction clearly describes prior studies, their limitations, and the motivation behind this work.
- The experiment design is convincing. It explored multiple dimensionality reduction methods and then used K-means clustering to identify clusters in the resulting low-dimensional space. The clusters were assessed with Kaplan-Meier curves for time to virological failure to determine if they differed by health outcomes. These analyses made the results convincing.

**Summary Of The Paper:**

The paper proposed a novel topic modeling method (Latent Dirichlet Allocation, LDA) to identify latent symptom clusters in a dataset consisting of one of the largest cohorts of people living with HIV (PWH). Several dimensional reduction approaches were investigated to extract meaningful latent information, and the traditional clustering method (K-means) was applied to the reduced dimension space to identify clusters of symptoms. The method outperformed traditional dimension reduction approaches such as NMF and PCA, as the clusters resulting from the proposed method showed a clearer association with the time to virological failure. Through the analysis, the paper identified a new cluster related to inflammatory responses, which could potentially serve as a prognostic marker of virological failure. Overall, this work could contribute to personalized symptom management and targeted interventions for PWH.

**Weaknesses:**

- The application of LDA+K-means on this domain seems novel, but the method itself (LDA) seemed traditional
- From the clustering results in Figure 2, I can clearly see co-occurrence patterns for symptoms. However, the SIR cluster (the cluster the paper claimed as novel) seemed quite weak compared to other clusters. Unlike other clusters, symptom probabilities are more spread out, and the gap between the 3rd and 4th symptoms is very small.
- Some descriptions of how health outcomes (time to virological failure) are defined were not clear. (See questions in the below section)
- In the Discussion/Conclusion:
    - The discovery of the SIR cluster would be more convincing if there were supporting clinical studies that observed similar patterns.

---

### Official Review · Reviewer_76gc · 2024-08-08
**This paper introduces an innovative approach to identifying symptom and patient clusters in individuals living with HIV (PWH) by adapting the Latent Dirichlet Allocation (LDA) method. The study is notable for its utilization of a large and reliable dataset, specifically the CNICS database, as well as being the first to apply topic modeling to identify clinically significant PWH symptom clusters associated with virological failure. While the application of LDA in this context is interesting, the method itself is not novel and has been used in other clinical research. The study demonstrates the potential of LDA for improving personalized treatment strategies for HIV patients, but its true innovation needs clearer articulation.**

**Overall Rating:** 5
**Confidence:** 4

**Other Quality Metrics:**

Clarity of writing: Good.
Clinical Significance: Good.
Methodological Novelty: Fair.
Experiments and Results: Good.

**Questions For The Authors:**

1. How would you further account for potential confounding variables like pre-treatment viral load and medication adherence in your analysis?
2. Can you provide more insights into the potential biological mechanisms behind the systemic inflammatory response cluster and its association with virological failure?
3. Given that LDA has been used in other clinical studies, what specifically is novel about your application of LDA to HIV symptom clustering?

**Strengths:**

1. While LDA is not a new method, the application of LDA to symptom clustering in HIV research is a novel contribution that expands the methodological toolkit available (currently only common dimensionality reduction techniques such as PCA, CFA, NMF) for analyzing clinical data.
2. The study uses a large and diverse cohort, enhancing the generalizability of the findings.
3. The identification of symptom clusters with significant prognostic value, particularly the SIR cluster, has potential clinical implications for personalized treatment strategies.
4. The study is methodologically sound, with a clear explanation of the dimensionality reduction techniques used and a robust validation of the results through clinical interpretability and statistical significance. Specifically, the author also explored various parameter settings for optimal performance for each methods, which enhanced the reliability of the new model.

**Summary Of The Paper:**

The study investigates a novel approach to identifying symptom clusters in individuals living with HIV (PWH) by utilizing data from the Centers for AIDS Research Network of Integrated Clinical Systems (CNICS).
The authors have adapted the Latent Dirichlet Allocation (LDA) model, commonly employed in text analysis, to uncover latent symptom clusters within a substantial cohort of PWH. These clusters are subsequently utilized for patient grouping and evaluation of their risk of virological failure.
Regarding the results, the research demonstrates that LDA surpasses traditional methods such as Principal Component Analysis (PCA) and Non-Negative Matrix Factorization (NMF) in discerning clinically relevant symptom patterns. The authors have identified five symptom clusters, including an innovative systemic inflammatory response (SIR) cluster strongly linked to an elevated risk of virological failure. The findings suggest that this approach has the potential to improve personalized symptom management and targeted interventions in HIV care.

**Weaknesses:**

1. LDA is a well-established method with existing applications in clinical research. For example, LDA has been used in studies to analyze patient symptoms in electronic health records and to identify disease subtypes. The innovation of applying LDA to HIV symptom data should be clearly articulated or with any updates to the methods, and the authors should clarify how this application advances the field.
2. The study relies heavily on self-reported data of patients outcomes, which can be subject to bias and affect the accuracy of the symptom clusters since different patients may have different standards of scoring their stress feeling.
3. The study does not address potential confounding factors such as pre-treatment viral load, medication adherence, and comorbidities, which could influence the results.

---

### Official Review · Reviewer_kjsC · 2024-08-28
**Review for a novol method in clustering HIV symptoms and patients**

**Overall Rating:** 7
**Confidence:** 3

**Other Quality Metrics:**

(a) Clarity of writing: good
(b) Clinical Significance: good
(c) Methodological Novelty: good
(d) Experiments and Results: good

**Questions For The Authors:**

N/A

**Strengths:**

1. The paper is well-organized, with clear language and structure that guides the reader through the study.
2. The paper presents a novel application of natural language processing to healthcare.
3. The results are well-discussed.

**Summary Of The Paper:**

The authors present a novel method for identifying symptom clusters in people living with HIV using a large dataset. The study leverages Latent Dirichlet Allocation (LDA), a topic modeling technique, to discover latent symptom clusters in this population. It then uses these clusters to categorize patients and predict their time to virological failure, which is a key health outcome in HIV treatment.

**Weaknesses:**

1. What does the input to your dimension reduction function look like besides LDA?
2. It would be helpful to see the clustering plot after kmeans and discuss the differences in clustering between different methods (LDA vs. traditional clustering method)

---

### Decision · Program_Chairs · 2024-09-23

Accept